

# Performance of post-processing algorithms for rainfall intensity measurements of tipping-bucket rain gauges

Mattia Stagnaro[1,2], Matteo Colli[1,2], Luca Giovanni Lanza[1,2], and Pak Wai Chan[3]

[1]University of Genova, Department of Civil, Chemical and Environmental Engineering, Via Montallegro 1, 16145 Genova, Italy
[2]WMO/CIMO Lead Centre "Benedetto Castelli" on Precipitation Intensity, Italy
[3]Hong Kong Observatory, 134A Nathan Road, Honk Kong, China

*Correspondence to:* Mattia Stagnaro (Mattia.Stagnaro@unige.it)

**Abstract.** A number of rain events recorded from May to September 2013 at the Hong Kong International Airport (HKIA) have been selected to investigate the performance of post-processing algorithms used to calculate the Rainfall Intensity (RI) from Tipping-Bucket Rain Gauges (TBRGs). We assumed a drop counter catching-type gauge as a working reference and compared rainfall intensity measurements with two calibrated TBRGs operated at a time resolution of 1 min. The two TBRGs

differ in their internal mechanics, one being a traditional single-layer dual-bucket assembly, while the other has two layers of buckets. The drop counter gauge operates at a time resolution of 10 s, while the time of tipping is recorded for the two TBRGs. The post-processing algorithms employed for the two TBRGs are based on the assumption that the tip volume is uniformly distributed over the inter-tip period. A series of data of an ideal TBRG is reconstructed using the virtual time of tipping derived from the drop counter data. From the comparison between the ideal gauge and the measurements from the two real TBRGs

the performance of different post-processing and correction algorithms are statistically evaluated over the set of recorded rain events. The improvement obtained by adopting the inter-tip time algorithm in the calculation of the RI is confirmed. However, by comparing the performance of the real and ideal TBRGs, the beneficial effect of the inter-tip algorithm is shown to be relevant for the mid-low range of rainfall intensity values (where the sampling errors prevail), while its role vanishes with increasing the RI, in the range where the mechanical errors prevail.

## 1 Introduction

Application-driven requirements of rainfall data (see e.g. Lanza and Stagi, 2008), the recommendations of international bodies such as the World Meteorological Organization (WMO, 2008), and new measurement quality standards issued at the national (UNI 11452:2012; BS 7843-3:2012) and international (CEN/TR 16469:2013) level provide an increasingly demanding framework in terms of proven instrumental accuracy and reliability.

Following the effort leaded in the last decade by WMO and aimed at quantifying the achievable accuracy of rainfall intensity measurements (Lanza and Vuerich, 2009), both users and manufacturers of precipitation gauges are developing strategies to reduce the uncertainty and to provide suitably documented performance evaluation of rainfall measurements.

Sound metrological procedures for the assessment of the uncertainty of meteorological measurements have been recently



introduced within the framework of Europe-wide collaborative projects (Merlone et al., 2015), and therein extended to the measurement of liquid precipitation (see Santana et al., 2015).

Beside the inherent instrumental factors (e.g. the systematic mechanical bias of tipping-bucket rain gauges, or the dynamic response bias of weighing gauges), post-processing of the raw data to obtain accurate rain intensity records at a pre-determined

temporal resolution is common practice in rain gauge measurements. In the case of tipping-bucket rain gauges (TBRGs), dedicated post-processing algorithms must be employed to achieve sufficient accuracy and to minimize the impact of sampling errors and the discrete nature of the measurement.

Various algorithms have been proposed to this aim, and discussed in the literature (Costello and Williams, 1991; Habib et al., 2001; Colli et al., 2013b, a). Yet, the operational practice of most users, including national weather services, still relies

on the trivial counting of the number of tips occurring in the desired time frame, and the rain depth per one minute (the WMO recommended time frame for rain intensity measurements) is obtained as the product of that number by the nominal volume of the bucket.

This method (as already observed by Costello and Williams, 1991) results in a general underestimation of rain intensity figures and in a high level of uncertainty, due to the random nature of the number of tips per minute within any real-world, highly

variable rainfall event. Moreover, the correction of systematic mechanical biases can not be optimized with this method since it would be applied to the averaged values only, and most tipping-bucket rain gauges show a nonlinear correction curve after laboratory calibration (Lanza and Stagi, 2009).

In this paper, the performance of different post-processing algorithms employed in the calculation of the rainfall intensity from tipping-bucked rain gauges are compared and discussed. Data recorded at a field test site by two TBRGs using different

mechanical solution are used, and a catching-type drop-counting gauge is assumed as the working reference. The comparison aims to highlight the benefits of employing smart algorithms in post-processing of the raw data, and their capability to improve the accuracy of rain intensity measurements obtained from TBRGs.

## 2  Field site and instrumentation

The Hong Kong Observatory performs rainfall measurements at the weather station of the Hong Kong International Airport

(HKIA). An Ogawa drop-counting rain gauge, model Osaka PC1122 (OSK), is available at the field site, providing rainfall measurements at the time resolution of 10 s.

Based on the calibrated drop size volume (calculated in October 2013) of 63.93 $mm^3$, the OSK drop counter rain gauge is able to measure rainfall rates up to 200 $mm\,h^{-1}$ with a resolution of $5.21\cdot10^{-3}$ mm (Chan and Yeung, 2004; Colli et al., 2013b), therefore fulfilling the WMO (2008) accuracy requirements.

Due to the high accuracy and time resolution we adopted the OSK drop counter gauge as a working reference, to be compared with co-located observations performed by two TBRGs manufactured by Logotronic MRF-C (LGO) and Shanghai SL3-1 (SL3). The main characteristics of the instruments employed in this study are summarized in Table 1. Note, in particular, that the two TBRGs have the same nominal sensitivity (equal to 0.1 mm), but employ different mechanical solutions. The LGO





is a traditional TBRG with a dual-compartment single-bucket assembly, while the SL3 gauge has two consecutive layers of dual-compartment buckets (see Figure 1).

This is not a common solution for TBRGs, and the objective of the two layers of buckets employed seems to reside in the attempt to reduce the systematic mechanical bias, typical of traditional TBRGs. In a sense, this is a hardware-type of correction

similar to the use of a syphon or other mechanical solutions.

In order to correct the systematic mechanical errors of the two TBRGs, both of them were subjected to appropriate dynamic calibration in the laboratory. The dynamic calibration consists of providing the gauge with a sufficient number of equivalent rainfall intensities, using calibrated constant flow rates. By comparing the reference values with those measured by the rain gauge under test, the parameters of a suitable correction curve (usually a power law) are derived. The measurements from the

two TBRGs were corrected before performing any comparison with the drop counter time series and/or with the ideal TBRG data obtained from the reference.

In this work, the computation of statistical estimators and deviations between paired observations was performed with no reference to any ancillary data (wind speed and direction, air temperature and absolute pressure, etc.) although it is known that some of them (especially the wind) may actually affect the accuracy of the measurement.

The field data available for this study cover a five-month period of observations from May to September 2013. Eight significant events in this period were selected on the basis of the total rainfall depth, after checking that the reference rainfall intensity values were lower than the given factory limits for the instruments under test.

Table 2 reports a short description of the selected events, in terms of total rainfall depth ($h_{tot}$), maximum rainfall intensity in one minute ($I_{max}$) and event duration ($d$).

Figure 2 shows a sample hyetograph of the raw data for the rain event occurred on 22 May 2013. The one-minute reference rainfall intensity is depicted (shaded gray background) as calculated from the 10 s high-resolution data of the OSK drop counter, as well as the cumulated rainfall for the OSK (solid red line) and the SL3 and LGO gauges (dashed and dotter line). The underestimation of the cumulative rain depth by the two TBRGs is evident from Figure 2. The relative underestimation of the total depth, for this particular event, is equal to 13.6% and 12.5% for the SL3 and LGO rain gauge respectively.

Figure 3 reports a non-parametric statistical description of the set of events in terms of one-minute rainfall intensity distribution from the reference DC (upper part), together with the total accumulation for the DC and the two TBRGs (lower part). Note that the two TBRGs show a larger underestimation of the total rain amount for events characterized by the highest values of the rainfall rate, in terms of both the mean and the extreme values, while the difference decreases for lower RI events.

## 3   Method

We adopted the catching-type drop counter gauge as the working reference for this work due to the high sensitivity of the measurement. Indeed, the instrument provides the number of generated drops with a time resolution of 10 s. We first aggregated this information to obtain the one-minute reference rainfall intensity values ($RI_ref$), for use in the overall assessment of the accuracy of the two involved TBRGs.



Both the SL3 and LGO rain gauges provide records of the time stamp of each tip. This feature allows using various algorithms to calculate the one-minute rainfall intensity values for the two investigated TBRGs.

The first, traditional and widely applied method to derive the one-minute rainfall intensity ($RI_raw$) simply relies on the counting of the number of tips within each minute. The product of this number by the nominal volume of the bucket provides

the rainfall amount in any single minute, and therefore the average rainfall intensity at such and any higher time resolution. Using a suitable correction curve derived from laboratory calibration allows accounting for systematic mechanical errors as a function of the rainfall intensity. The traditional method assigns the whole volume of each bucket to the minute in which the tip occurs, even when part of the bucket is actually filled already in the previous minute, introducing significant counting errors in the calculation of the one-minute rainfall intensity $RI_raw$. The uncertainty introduced by the tip counting method also affects

the efficacy of the calibration, since the correction applied to the volume of the bucket in each minute does not precisely derive from the actual rain intensity occurring in that minute.

The second method used to obtain the one-minute rainfall intensity values ($RI_{Ttip}$) employs the inter-tip time algorithm (see e.g., Costello and Williams, 1991; Colli et al., 2013b), which is based on the assumption that the nominal volume of each bucket is equally distributed over the inter-tip period. The calculation of the $RI_{Ttip}$ for each minute accounts for the portion of

the inter-tip period actually falling into that minute. In this way, also the calibration is the most effective since the correction applied to the volume of the bucket at the variable inter-tip scale is precisely the one corresponding to the measured rainfall intensity.

For both the TBRGs, the two values of one-minute $RI$ derived from the two post-processing algorithms described above (generally indicated here as the measured rain intensity $RI_m$) were employed to calculate the accuracy of rainfall intensity

measurements in terms of deviations from the DC reference value as follow:

$$e_{rel}(\%) = \frac{RI_m - RI_{ref}}{RI_{ref}} \cdot 100 \qquad (1)$$

In addition, in order to compare the performance of post-processing and correction algorithms for the TBRG measurements, we derived a virtual sequence of tips of an ideal TBRG from the high-resolution DC data. This simulates the behavior of a best performing TBRG, representing the maximum accuracy to be expected when using a tipping-bucket mechanics. The inter-tip

algorithm was employed to derive the one-minute ideal rainfall intensity values ($RI_{ideal}$).

In order to assess the capability of the employed algorithm to describe the inner variability of the considered events and to capture their finer details, we calculated the correlation coefficients between all the derived time series and the reference ones. In particular, for each event, we calculated the RMS error of paired deviations between the measured and the ideal/reference $RI$ signal.

In an effort to homogenize the dataset, we considered the normalized value of the generic rainfall intensity value $RI$ obtained as follow:

$$RI_n(\%) = \frac{(RI - \mu_{ref})}{\sigma_{ref}} \qquad (2)$$





where the mean value $\mu_{ref}$ and the standard deviation $\sigma_{ref}$ employed to obtain the normalized values of $RI$ for all the TBRGs, are those derived from the drop counter reference rainfall rates. In this way, we obtained comparable results in terms of standard deviation of the normalized $RI$ time series for all the investigated rainfall events.

## 4   Results

We first evaluated the accuracy of the investigated TBRGs by comparing their performance with the working reference. Figure 4 shows the relative deviations ($e_{rel}$) from the reference DC for the three TBRGs (including the ideal virtual gauge). The reported boxplots provide a synthesis of the results obtained adopting the inter-tip algorithm and are classified according to different ranges of $RI_{ref}$.

Two different regions of this graph show different behavior of the relative deviations: for low values of the $RI$ the relative
deviations of all the TBRGs exhibit a large variability, whereas this scatter suddenly decreases just above the $RI$ value of 6 mm h$^{-1}$ (highlighted in black) and then continues to reduce with increasing the $RI$. This limit coincides with the sensitivity of the TBRGs (both the real and virtual ones); in fact, values of RI higher than 6 mm/h generate at least one tip per minute for TBRGs with a sensitivity of 0.1 mm.

It emerges from the graph in Figure 4 that the average deviation of both the LGO and SL3 gauges is always negative. In
particular, the SL3 gauge shows an average underestimation that is larger than the LGO (except for one bin). This is coherent with the daily amounts shown in Figure 3, where the cumulative rain depth of the SL3 gauge is lower than or equal to the value recorded by the LGO. Despite the SL3 rain gauge underestimates rainfall more than the LGO on average, the variability of the deviations from the reference for different values of RI is slightly reduced with respect to the LGO.

The behavior of the ideal TBRG is clearly different, since it is not affected by instrumental mechanical errors (ideal mechanics),
and the average value of $e_{rel}$ becomes close to zero immediately after the threshold value of the instrument sensitivity.

Note that the ideal TBRG shows a large variability and an average value of the relative deviations that is comparable to the real TBRGs when the RI values are below 6 mm h$^{-1}$. This means that the error caused by the aggregation time is relevant in this region when compared to the mechanical one.

As the rainfall rate increases, the variability of $e_{rel}$ considerably decreases, even if does not vanish due to the sampling time of
the DC (10 seconds), whereas the average values are close to zero.

Figure 5 describes the relative deviation ($e_{rel}$) from the DC reference of the measured value of RI from the three TBRGs adopting the rough tip counting algorithm. Below the instrument sensitivity limit of 6 mm h$^{-1}$, where less than one tip per minute occurs, the relative errors of all TBRGs are similar and show a very high scatter. Above this threshold value, the variability gradually decreases. Also in this case the ideal TBRG shows an average value of deviation which is close to zero,
while the SL3 and LGO continue to maintain a mean value of $e_{rel}$ negative for all the RI classes . Close to the threshold value of 6 mm h$^{-1}$, the variability of the relative error ($e_{rel}$) is drastically reduced for all three gauges, and the average value is the closest to zero.



In Figure 6 the Taylor diagram is depicted (Taylor, 2001) to show the effect of the inter-tip algorithm in terms of the standard deviation of the normalized RI signal, the correlation coefficient between the TBRGs and the reference and its deviations from the reference. Considering the standard deviation of the RI signal for the real TBRGs, note that the two algorithms used to compute the RI values provide comparable results (approximately equal to 0.9). Since the ideal TBRG directly derives from the reference, the normalized standard deviation is the closest to unity.

In the same figure, the benefit of using the inter-tip time algorithm instead of the tip counting is evident: the correlation coefficients of the two real TBRGs increase using the former one for both TBRGs. Therefore, the beneficial effect in terms of deviations of the measured RI from the reference is highlighted. In fact, note the reduction of the RMS difference from the reference, approaching the value of the ideal TBRG. This reduction can be quantified in about 1 mm h$^{-1}$.

Also, the RI time series of the ideal TBRG show a normalized standard deviation approximately equal to 1, that is the same as the reference, and a mean correlation coefficient greater than 0.99. However, the average value of the RMS between the synthetic TBRG and the reference continues to show a relevant value slightly below 2 mm h$^{-1}$, which does not decreases under the value of 1 mm h$^{-1}$ in all the considered events.

In order to evaluate the effectiveness of the post-processing algorithms on the accuracy of the measurements over different ranges of rainfall intensity, we plotted the standard deviation of the relative error ($e_{rel}$) for different classes of RI. Figure 7 reports the results of this analysis.

It is evident from the graph that the raw counting of the number of tips results, for both the investigated TBRG, in a continuous trend of linear (in a log-log scale) reduction of the error variance with increasing the RI. This reflects the fact that the random attribution of one tip to the wrong minute does not affect much the derived RI since the number of tips per minute is relatively high.

At very low values of RI, there is little difference between the results obtained by employing the simple counting of tips or the inter-tip algorithm, with respect to the ideal TBR. By increasing the RI, but still below the threshold intensity corresponding to the sensitivity of the gauges, the effectiveness of the inter-tip time algorithm is relevant and results are very near to the ideal gauge. This effectiveness decreases beyond the sensitivity value and, with increasing the RI beyond 50-60 mm h$^{-1}$, the difference with respect to the counting of tips becomes negligible. In this range, the LGO performs slightly better than the SL3 when the inter-tip time algorithm is used.

## 5   Conclusions

The raw data recorded during a dedicated monitoring campaign have been analysed using two different post-processing algorithms to calculate the one-minute RI series. The results allow comparing the performance of the inter-tip time algorithm with the more common tip counting method, when using two different types of TBRGs. The field reference chosen for this comparison is a catching type, optical drop counter that, although calibrated in the laboratory, is still subject to unknown uncertainties in field operation. Notwithstanding this residual uncertainty, comparison of the two gauges with a virtual TBRG obtained from



the reference measurements was able to show relevant differences in the calculated rainfall intensity, and the relation of such differences with the rainfall rate itself.

In particular, the main benefit of adopting the inter-tip time method as a post-processing algorithm to calculate rainfall intensity from the raw data resides in a better representation of the inner variability of rainfall events. The measured RI series

shows an improved correlation coefficient and a lower RMS with respect to the reference, closely approaching the performance of an ideal TBRG, which is not affected by mechanical biases.

In terms of accuracy, the inter-tip time algorithm contributes its greater beneficial effects in the range of low to mid RI values. In the very low RI range, below the threshold value of $6 \ \mathrm{mm \ h^{-1}}$ (corresponding to the sensitivity of the involved instrument and a typical value for operational TBRG) the performance of the inter-tip time method in terms of the statistical amplitude of

10 the deviations from the reference of the calculated RI are comparable to an ideal TBRG. In this range, indeed, the time period between consecutive tips exceeds the time resolution of the measurement, and the sampling error represents the main source of uncertainty.

Beyond that threshold value, a step change is observed since at least one tip per minute is recorded when $\mathrm{RI} > 6 \ \mathrm{mm \ h^{-1}}$. The inter-tip time algorithm is still better than the tip counting method in this range, up to about $50 \ \mathrm{mm \ h^{-1}}$, although the

15 performance are no more comparable to the ideal gauge. Mechanical errors become prevalent here, so that deviations from the ideal gauge result from the residual uncertainty of the calibration process. At the highest RI values the benefits of the inter-tip time algorithm vanish due to the high number of tips per minutes recorded in this range, and the performance of the two post-processing algorithms become comparable, though both of them perform worse than the ideal TBRG.

Data availability: the data presented in this paper are available on request from the corresponding author.

*Acknowledgements.* This research is supported through funding from European Metrology Research Programme MeteoMet 2 (ENV58-REG3).





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



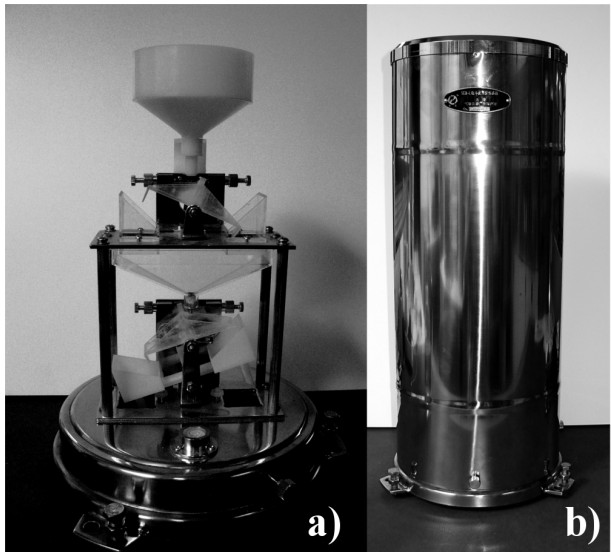

**Figure 1.** Shanghai SL3-1 (SL3): internal mechanism with double layer of tipping buckets (a) and external case (b).

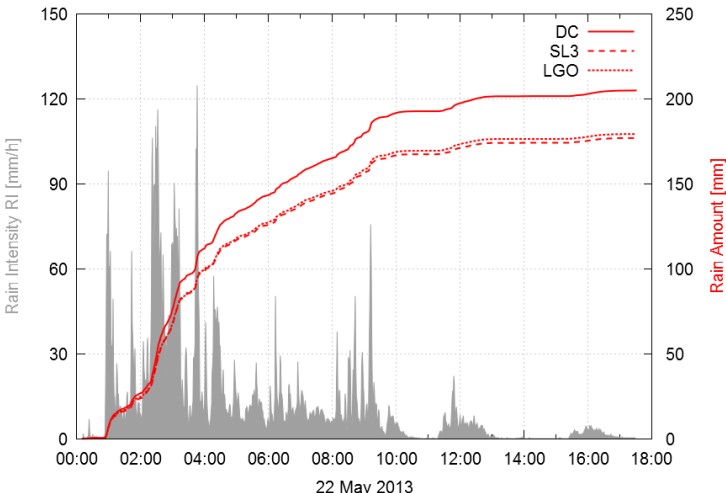

**Figure 2.** Reference rainfall intensity measured by the Ogawa drop counter (shaded gray background) and comparison of the cumulated reference with the cumulated Logotronic (LGO) and Shanghai (SL3) measurements during the sample event of 22 May 2013.





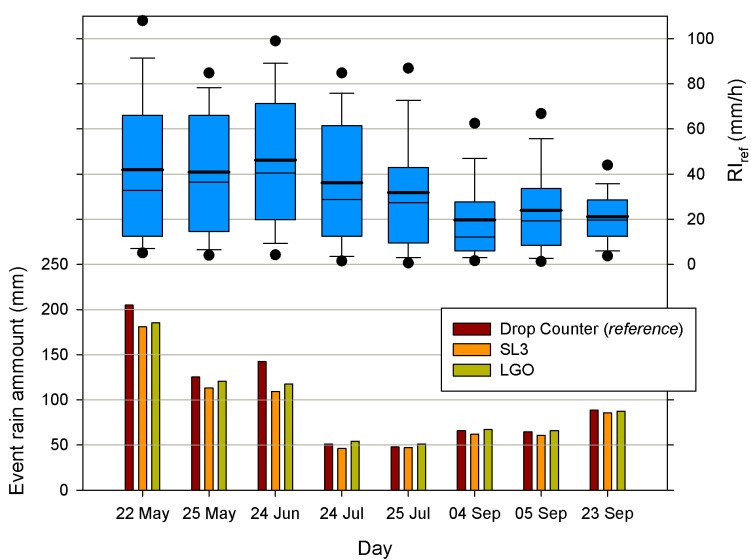

**Figure 3.** Non-parametric distribution of the reference rainfall intensity for each event and corresponding daily rain amount for the Drop Counter (reference) and the two TBRGs (SL3 and LGO).

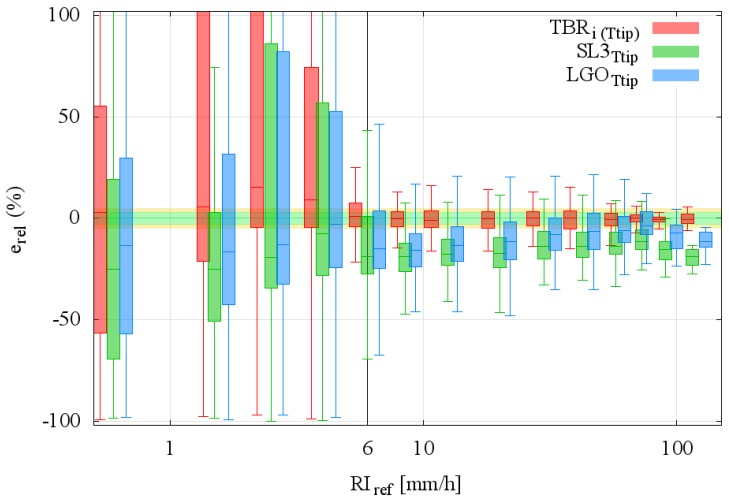

**Figure 4.** One-minute relative deviations ($e_{rel}$) between the three TBRGs (including the ideal one) and the reference (DC) when adopting the inter-tip post-processing algorithm.




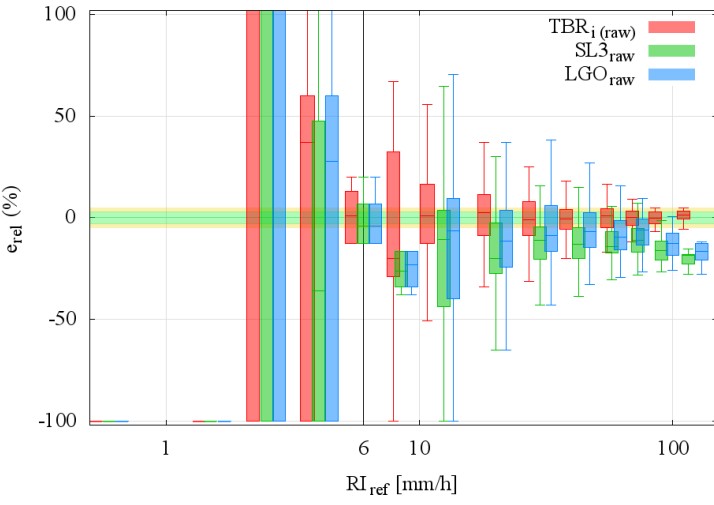

**Figure 5.** One-minute relative deviations ($e_{rel}$) between the three TBRGs (including the ideal one) and the reference (DC) when adopting the tip counting algorithm.

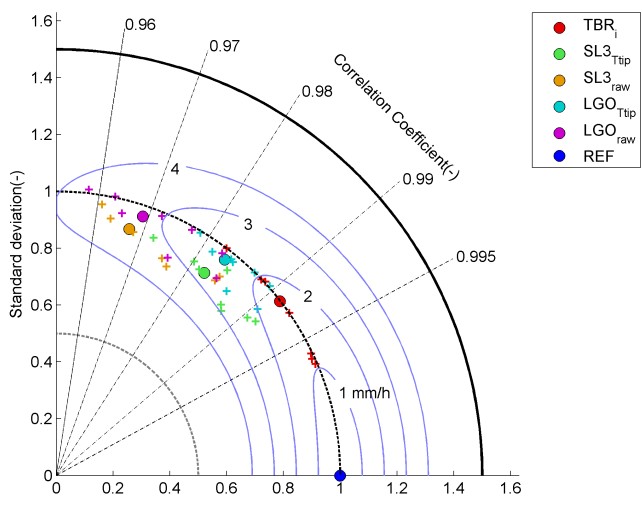

**Figure 6.** Taylor diagram representation of pattern statistics of the various RI series. The radial distance from the origin is proportional to the normalized standard deviation of the RI signal; the blue contour lines highlight the RMS difference from the reference (blue dot) for each recorded event; the azimuthal position indicates the correlation coefficient between the RI signal and the reference. Crosses indicate the statistics of each single event, while the dots indicate the average values of the whole campaign (colors according to the legend).



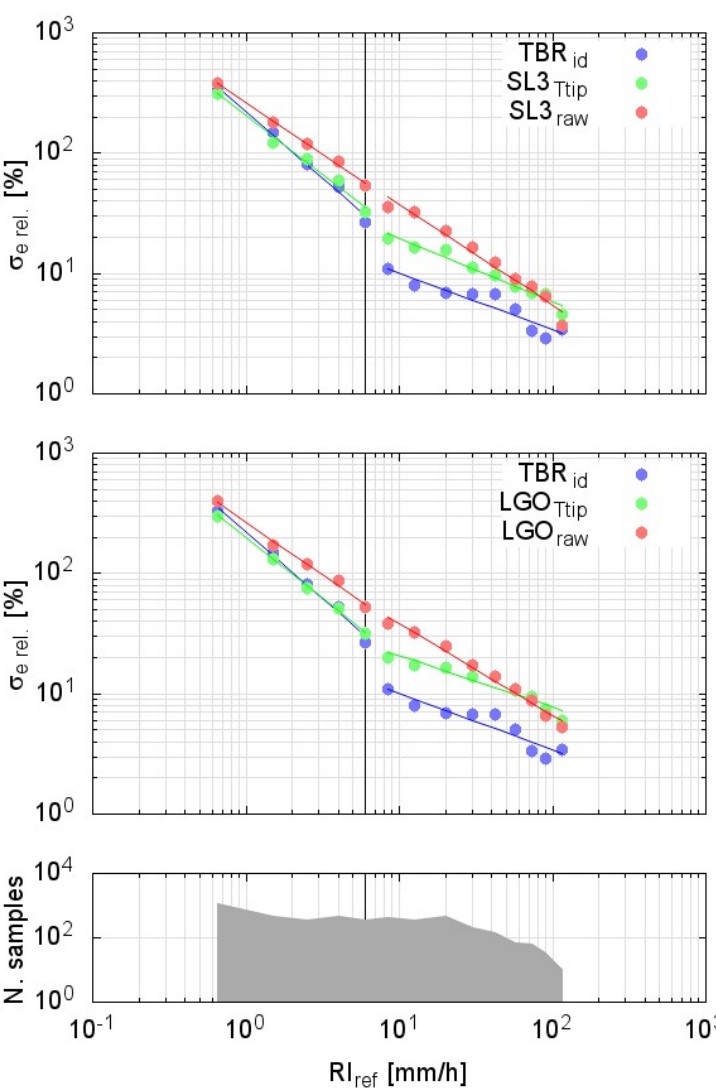

**Figure 7.** Standard deviation of the relative error for the ideal and the investigated TBRGs when adopting the inter-tip time algorithm and the tip counting method with respect to the ideal gauge for both the SL3 (a) ad LGO (b) instruments.



**Table 1.** Type of rain gauges employed for this comparison and their principal characteristics.

| Rain gauge | Measuring principles | Resolution | Maximum rainfall intensity* | Funnel diameter |
|---|---|---|---|---|
| | | (mm) | (mm h$^{-1}$) | (mm) |
| Ogawa OSK PC1122 | Drop counter | $5.21 \cdot 10^{-3}$ | 200 | 127.0 |
| Logotronic (LGO) | Tipping-bucket | 0.1 | 200 | 252.3 |
| Shanghai (SL3) | 2-layers tipping-bucket | 0.1 | 240 | 200.0 |

*Maximum measured rainfall intensity as declared by the manufacturer



**Table 2.** Total rainfall depth ($h_{tot}$), maximum one-minute rainfall rate ($I_{max}$) and duration ($d$) of selected events recorded by the Ogawa drop counter during the observation period.

| **Date** | $h_{tot}$ | $I_{max}$ | $d$ |
|---|---|---|---|
| | (mm) | (mm h$^{-1}$) | |
| 22 May 2013 | 205 | 125 | 11h 27' |
| 25 May 2013 | 125 | 102 | 06h 56' |
| 24 June 2013 | 142 | 114 | 06h 19' |
| 24 July 2013 | 51 | 95 | 02h 54' |
| 25 July 2013 | 48 | 96 | 03h 10' |
| 04 September 2013 | 66 | 72 | 06h 44' |
| 05 September 2013 | 65 | 73 | 04h 53' |
| 23 September 2013 | 89 | 64 | 06h 09' |