# Peer review of "Performance of post-processing algorithms for rainfall intensity measurements of tipping-bucket rain gauges"

_Atmospheric Measurement Techniques, 2016_

## Referee Comment (RC1) · Anonymous Referee #1 · 15 Aug 2016

The manuscript represent a substantial contribution to scientific progress, it addresses an important topic involving the processing of data of liquid precipitation using a set of different rain gauges. The results help researchers and technologists in making the decision for possible updates of the measurement systems and statistical treatment of data of liquid precipitation (rain). The manuscript is well-organized. The sections are well-developed requiring some revisions. The authors answer the questions that they set out to answer. The methodology is clearly explained, but requiring minor revision. The author's results are convincing because reports detailing the performance of algorithms with temporal resolution more precise based on measures of different rain gauges. The title could be "Performance of post-processing algorithms for rainfall

intensity using measurements of tipping-bucket rain gauges" or "Performance of post-processing algorithms for rainfall intensity using measurements of observation system with tipping-bucket rain gauges" because TBRGs measure the amount of rain (precipitation liquid) and determination of RI is calculated using a system (software). In the "1 Introduction" section: - a) it is interesting to describe the term "measurement uncertainty" according to the VIM (International Vocabulary of Metrology – Basic and General Concepts and Associated Terms). In the "2 Field site and instrumentation" section: - a) page 2, line 25 include a better description of the OSK, such as operating principle (optical), technical characteristics (catchment area, accuracy of measurement, nominal uncertainty, etc.) and the calibration date (corrections, expanded uncertainty, coverage factor, confidence level, etc.); - b) page 2, line 32, in Table 1, include more information on the manufacturers' specifications (catchment area, accuracy of measurement, nominal uncertainty – if provided, etc.); - c) page 3, describe the installation of rain gauges (height, distance between them, obstacles, etc.). These factors may be relevant sources of uncertainty of measurements. If possible include figure of site; - d) page 3, describe the complete system from the rain gauge (measurement) until the final data. The TBRGs measures the amount of rain, it means that there is a data acquisition system for storing the tips and timestamp (hh:mm:ss) and a system for calculating the RI. How the RI was obtained (datalogger, software)?; - e) page 3, was calibration performed for the complete system (sensors and data acquisition system or only sensors)? The datalogger's contribution may be insignificant, but what is the uncertainty of the datalogger in the time record (timestamp)? Describe uncertainty or accuracy of datalogger. It may be important to one of the algorithms; - f) page 3, report data from the "Calibration Certificate" / calibration (correction, uncertainty, etc.) of all sensors and range. Instruments with different measuring principles produce different results, but compatible in most cases (statistically equal), depending on their measurement uncertainties; - g) abbreviations or acronyms used in figures should be explained in the legend (see figure 2 and review all figures). In the "3 Method" section: - a) review all acronyms, for example

RIref and not RIref, RIraw and not RIraw (see attached revision pdf), etc, including the graphics and legends of graphics and figures (see the "Manuscript composition" of Manuscript preparation guidelines for authors of AMT); - b) use the same acronym for reference (DC or OSK or REF?) in the text, graphics and the chart legends, as well as other sensors; - c) Figure legends should clarify all the symbols used and should appear in the figure (see the "Manuscript composition" of Manuscript preparation guidelines for authors of AMT); - d) Use the term uncertainty, as described in VIM; - e) Describe all terms used in the formulas, for example RIn (and which values for n?, review all formulas); - f) page 4, equations should be referred to in the text according to the Manuscript preparation guidelines for authors of AMT; - g) What were the computational tools (language, software, operating system) used to run programs (algorithms under study) and how RI is calculated using the TBRG LGO, SL3 and OSK/REF? In the "4 Results" section: - a) page 5, the TBRG measures the amount (tips) of rain. The limit is the limit of the system that calculates the RI; - b) and how the propagation of uncertainty for calibration of the sensors was treated in the results? It is interesting to express "the measure $\pm$ uncertainty" for each rain gauge; - c) due to the graphic resolution; it is interesting to show clearer in the chart when the exception occurs; - d) the abbreviation "Fig." should be used when it appears in running text and should be followed by a number unless it comes at the beginning of a sentence, e.g.: "The results are depicted in Fig. 5. Figure 9 reveals that..." (see the "Manuscript composition" of Manuscript preparation guidelines for authors of AMT); - e) The TBRi(Ttip) acronym is not commented in the text and in the figure legend; - f) page 5, line 21, list the figure of this paragraph; - g) improve the Y axis scale of graphic of figure 5 (review all scales of all graphics). In the "5 Conclusions" section: - a) page 6, line 31, cite examples of sources of uncertainties in field operation; - b) rewrite the paragraph to make the text clearer, once the rain intensity is calculated from the measured rain (tips) made by TBRGs and the time resolution used is usually 1 minute, but other time resolutions may be chosen. Make it clear when compare results obtained by different forms of RI calculations.

Please also note the supplement to this comment:
http://www.atmos-meas-tech-discuss.net/amt-2016-257/amt-2016-257-RC1-
supplement.pdf

―――――――――――――――――

---

## Referee Comment (RC2) · R. Uijlenhoet (Referee) · 16 Aug 2016

This paper deals with a comparison of two post-processing algorithms for tipping bucket rain gauges. The analysis is based on eight rainfall events collected in Hong Kong. Two algorithms are compared, one based on the number of tips per time interval and another based on the inter-tip time interval. An optical drop-counting rain gauge with a 10 sec time resolution is used as reference. The intercomparison is carried out at a time step of 1 min. Both tipping bucket rain gauges have a nominal volume resolution of 0.1 mm, which means that a tip frequency of one per minute corresponds to a rain rate of 6 mm per hour.

The main conclusion of the paper is that the algorithm based on inter-tip times outper-

forms the algorithm based on tip counts, except for very small rain rates (< 0.6 mm/h), where the performance of both algorithms becomes nearly the same. Up to a rain rate of 6 mm/h the performance of the inter-tip algorithm is found to be nearly equal to that of an artificial "ideal" tipping bucket rain gauge, simulated on the basis of the optical drop-counting gauge data. Beyond that, the inter-tip algorithm still outperforms the tip-counting algorithm, but its performance is less than that of the ideal gauge. The authors convincingly show that sampling errors play a major role for low to intermediate rain rates, whereas mechanical errors play a dominant role for larger rain rates. The latter lead to underestimation of rainfall accumulations for events containing intervals with such large rain rates (Fig. 2).

Tipping bucket rain gauges are the standard in many applications in meteorology and hydrology. Therefore, a clear understanding of their performance and the sources of error involved is of significant practical and scientific relevance. As such, the topic of this paper fits AMT well. The first reviewer has a number of editorial remarks, which I largely agree with. I have attached an annotated version of the manuscript for the authors to improve their paper for resubmission.

A few remarks added to the annotated manuscript are repeated here. First of all, I am not convinced Fig. 6 (the Taylor diagram) is the best way to present the results of the intercomparison. In any case, it was not easy for me to decipher it (particularly the choice of colors for the symbols was confusing to me). Also, the "residual uncertainty in the calibration process" (see the Conclusions section) asks for further discussion. Are there ways to account for this remaining error source in the calibration process? It is an important issue, because we are not only talking about uncertainty (random error), but also about bias (systematic error), in this case underestimation (see Fig. 2), which can have important consequences in (meteorological and hydrological) applications.

Please also note the supplement to this comment:

http://www.atmos-meas-tech-discuss.net/amt-2016-257/amt-2016-257-RC2-supplement.pdf

**Supplement:**

[revised manuscript text omitted]

---

## Referee Comment (RC4) · Anonymous Referee #4 · 13 Sep 2016

In this paper, the performances of two different post-processing algorithms employed in the calculation of the rainfall intensity from tipping-bucked rain gauges (TBRGs) are compared and discussed. Data recorded at a field test site by two TBRGs using different tipping bucket assemblies are used, and a catching-type drop-counting gauge is used as the working reference. The comparison demonstrates the benefits, in terms of improved accuracy, of employing an inter-tip algorithm to compute intensities rather than the simpler and widely used procedure of counting tips within the time intervals. The paper is well written and the results are clearly explained. The main conclusion drawn by the authors is well supported by the evidence they have provided, namely that the inter-tip algorithm provides a much better basis for deriving intensity data than

simply counting the tips within the time intervals. The paper should therefore be published, but would benefit from some discussion of the points raised below, and some minor corrections. In particular, the characteristics of the errors, as summarized by the relatively large observed biases and variabilities relative to the reference measurements, merits some discussion. One of the referees has commented on the need for a site description which should include the spatial layout/separation distances/mounting heights of the gauges etc that might influence the characteristics of the errors. Given that the TBRGs are stated in the paper to have been subjected to laboratory calibration/correction of the intensity data using a correction curve, it does not seem that the errors observed in relation to the reference raingauge can be attributed to 'instrumental mechanical errors', given that the correction curve is supposed to account largely for these effects. Small scale rainfall variability across the measurement site must make a contribution to the observed variability. In relation to the consistent underestimation observed, which should be largely removed by the correction curve, was a volumetric check gauge run alongside the two TBRGs and the reference gauge to check on the total volumes from each over the period of the experiment? The authors are invited to discuss these points and to clarify them, given their extensive experience of conducting laboratory and field experiments on the performance of TBR rain gauges.

Some minor points: P1, L20: 'Following the effort led….' P1, L23: Uncertainty can be defined in a number of ways. The authors should explain early on how they are going to quantify the accuracy of their measurements. P2, L5: 'in rain gauge measurements'. Suggest eliminating this as it is redundant P2, L8: change to ' …to achieve this aim…' P3,L2: 'This is not a common solution for TBRGs……' Suggest no new para here as you are continuing the discussion of the dual layer assembly. P3,L6: '….systematic mechanical errors…' Suggest a brief explanation of how they arise. P3,L8-11: was the correction curve based on data derived from interval data or inter-tip data? Based on what follows in the paper, the former would appear to be preferable.

P3,L23: replace 'cumulated' with 'accumulated' throughout. P3,L24: change to '…

gauges respectively.' P3,L24: '. . .13.6% and 12.5% for the SL3 and LGO rain gauges respectively.' Presumably, these underestimation errors are after the correction curve referred to above has been applied. They are therefore quite large, so can they just be ascribed to systematic mechanical errors? P4,L1: change to '. . .allows various algorithms to be used. . .' P4,L6: change to '. . ..allows systematic mechanical errors to be accounted for. . ..' P4,L22-24: a brief description of how the ideal series of tips was derived would be helpful. P4,L24: change to '. . ..when using tipping-bucket mechanics. . .' P5,L1:'normalized' is a term used to describe a transformation to a normal distribution. Better to use 'standardized' P5,L21-22: the ideal TBR generally exhibits a significant positive bias for values in the range 0-6 mm which is different from the other two TBRs – reason for this? Figure 2.. Replace 'cumulated' with 'accumulated.(twice) Figure 3. It would be good to remind the reader what the bounds are in the box-and-whisker diagram

---

## Author Comment (AC1) · 25 Oct 2016

Matteo Colli[1,2], Mattia Stagnaro[1,2], Luca G. Lanza[1,2], and P.W. Chan[3]

[1]University of Genova, Department of Civil, Chemical and Environmental Engineering, Via Montallegro 1, 16145 Genova, Italy
[2]WMO/CIMO Lead Centre "Benedetto Castelli" on Precipitation Intensity, Italy
[3]Hong Kong Observatory, 134A Nathan Road, Honk Kong, China
*Correspondence to:* Mattia Stagnaro (Mattia.Stagnaro@unige.it)

Let us first thank Reviewer #1 for his/her constructive comments and the careful scrutiny of our paper that certainly improved the quality of our manuscript. Below we provide a point-by-point reply to the issues raised in the review (here reported in bold), while in a separate comment, a revised version of the manuscript is presented.

**The title could be "Performance of post-processing algorithms for rainfall intensity using measurements of tipping-bucket rain gauges? or "Performance of post-processing algorithms for rainfall intensity using measurements of observation system with tipping-bucket rain gauges" because TBRGs measure the amount of rain (precipitation liquid) and determination of RI is calculated using a system (software).**

We accepted the reviewer suggestion and changed the title as follow:

"Performance of post-processing algorithms for rainfall intensity using measurements from tipping-bucket rain gauges."

**In the "1 Introduction" section: - a) it is interesting to describe the term "measurement uncertainty" according to the VIM (International Vocabulary of Metrology - Basic and General Concepts and Associated Terms).**

In accordance to the reviewer, we describe the term "measurement uncertainty", following the VIM (JCGM, 2012) definition, in the introduction of the revised version of the manuscript.

**In the "2 Field site and instrumentation" section: - a) page 2, line 25 include a better description of the OSK, such as operating principle (optical), technical characteristics (catchment area, accuracy of measurement, nominal uncertainty, etc.) and the calibration date (corrections, expanded uncertainty, coverage factor, confidence level, etc.);**

As the referee pointed out, the operating principle of OSK is optical, this has been clarify in the text.

Concerning the technical characteristics, the catching area is already reported in Table 1 (expressed by the funnel diameter). In this table, we added the values of measurement accuracy for the OSK and the two TBRGs.

Regarding the calibration of the OSK, we now indicate the uncertainty of the drop volume, which is declared as $63.93 \pm 1.9$ $mm^3$ (in the range from 10 to 120 $mmh^{-1}$), which translates into a rainfall depth sensitivity of $5.21 \pm 0.15 \cdot 10^{-3}$ mm.

**b) page 2, line 32, in Table 1, include more information on the manufacturers' specifications (catchment area, accuracy of measurement, nominal uncertainty - if provided, etc.);**

The information about the collector's area was already present in Table 1 (expressed as the diameter of the instrument funnel) and we added in Table 1 the measurement accuracy of the TBRGs as provided by the manufacturers. No further information is available from the manufacturers.

**c) page 3, describe the installation of rain gauges (height, distance between them, obstacles, etc.). These factors may be relevant sources of uncertainty of measurements. If possible include figure of site**

We included in this section a description of the field test site and a drawing where the position of the three instruments is highlighted, as follows:

" *The three instruments have been placed in the west corner of the field test site of the Hong Kong Observatory (depicted in Fig. 1). Both the TBRGs and the OSK drop counter have been installed on the ground. The minimum distance from the main obstacle close to the instruments is about 18 m. From Fig. 1 it is possible to observe the position of the SL3 (blue box), LGO (red box) and the OSK (green box); the distance between the SL3 and the OSK is about 5.8* m, *while the LGO is located at about 2.1* m *south of the OSK.*"

[Figure]

**Figure 1.** The West corner of the Hong Kong Observatory field test site where the SL3 (blue box), the LGO (red box) and the OSK (green box) are located. The distance of each instrument from the field site borders is indicated.

**d) page 3, describe the complete system from the rain gauge (measurement) until the final data. The TBRGs measures the amount of rain, it means that there is a data acquisition system for storing the tips and timestamp (hh:mm:ss) and a system for calculating the RI. How the RI was obtained (datalogger, software)?;**

For the TBRGs, each instrument is simply connected with a data-logger and the timestamp is recorded whenever a tip occurs. Regarding the OSK drop counter the data-logger records the number of detected drops every 10 seconds. The timestamps are

stored in the form "yyyy-mm-dd HH:MM:SS.000". For this study, the RI is calculated via software starting from the timestamps recorded by the data-logger.

**e) page 3, was calibration performed for the complete system (sensors and data acquisition system or only sensors)? The datalogger's contribution may be insignificant, but what is the uncertainty of the datalogger in the time record (timestamp)? Describe uncertainty or accuracy of datalogger. It may be important to one of the algorithms;**

The calibration was performed only for the sensors and not for the data-logger. The sensitivity pf the data-logger is 1/1000 s, while the associated uncertainty is unknown. The uncertainty of the data-logger is not expected to influence the results significantly.

**f) page 3, report data from the "Calibration Certificate" / calibration (correction, uncertainty, etc.) of all sensors and range. Instruments with different measuring principles produce different results, but compatible in most cases (statistically equal), depending on their measurement uncertainties;**

Calibration was performed for all the instruments. For the OSK drop counter the drop volume has been provided to indicate the precision level of the instrument. For TBRGs a power law correction curve has been applied to the dataset of each instruments. Since the aim of this work consists in comparing the use of two RI algorithms, we have not indicated the parameters of those curves and we have not shown the beneficial effects derived from adopting the correction curves.

**g) abbreviations or acronyms used in figures should be explained in the legend (see figure 2 and review all figures).**

Indeed, there were inconsistencies in the use of some acronyms and abbreviations in the submitted manuscript. We checked the text to fix it, and the legend of each figure has been also checked in order to avoid any unexplained terminology.

**In the "3 Method" section: - a) review all acronyms, for example $RI_{ref}$ and not $RI_ref$, $RI_{raw}$ and not $RI_raw$, etc, including the graphics and legends of graphics and figures (see the "Manuscript composition" of Manuscript preparation guidelines for authors of AMT);**

Acronyms have been checked throughout the manuscript, including captions and legends of each figure, according to the "Manuscript composition" guidelines of AMT.

**b) use the same acronym for reference (DC or OSK or REF?) in the text, graphics and the chart legends, as well as other sensors;**

We checked the manuscript and changed the dual acronyms for all the sensors in the revised version of the manuscript.

**c) Figure legends should clarify all the symbols used and should appear in the figure (see the "Manuscript composition" of Manuscript preparation guidelines for authors of AMT);**

Following the referee comment, the legends of the figures have been verified and modified in order to explain all the symbols used.

**d) Use the term uncertainty, as described in VIM;**

We rephrased using the term *inaccuracies* rather than *uncertainty* since it seems more appropriate in the context of this sentence.

**e) Describe all terms used in the formulas, for example $RI_n$ (and which values for n?, review all formulas);**

We clarified in the revised version of the manuscript the terms used in the mentioned equation.

Regarding the terms $RI_n$, the subscript "n" indicates a "standardized" value of the one-minute rainfall intensity variable, regardless of the instrument and the algorithm used.

**f) page 4, equations should be referred to in the text according to the Manuscript preparation guidelines for authors of AMT;**

All equation references have been corrected following the Manuscript preparation guidelines.

**g) What were the computational tools (language, software, operating system) used to run programs (algorithms under study) and how RI is calculated using the TBRG LGO, SL3 and OSK/REF?**

The software used to manage data was developed by the authors using Matlab codes in the MS Windows operating system. The one-minute values for RI are calculated starting from the timestamp of the tips of the TBRGs and the drop frequency for the OSK. The calculation for the OSK consists in a simple count of droplets falling in each minute. Regarding the TBRGs, for the tip-counting algorithm the calculation of RI consists in a simple count of tips for each minutes, while for the inter-tip algorithm the one-minute RI values have been calculated performing a weighted average of the inter-tip RIs falling in the same minute.

**In the "4 Results" section: - a) page 5, the TBRG measures the amount (tips) of rain. The limit is the limit of the system that calculates the RI;**

We adjusted the text accordingly in order to separate the effect of the RI calculation method from the gauge characteristics.

**b) and how the propagation of uncertainty for calibration of the sensors was treated in the results? It is interesting to express "the measure $\pm$ uncertainty" for each rain gauge;**

The propagation of the calibration uncertainty was not considered in the results. The objective of this paper is not to investigate the sources of uncertainty and their contribution, but rather to compare two different RI calculation methods by using field measurements. We added a simple comment to section 4, when commenting Figure 5, to clarify this point.

**c) due to the graphic resolution; it is interesting to show used clearer in the chart when the exception occurs;**

We highlight in the graphs the limit of $6 \ \mathrm{mmh^{-1}}$ where the behaviors of the algorithms change.

**d) the abbreviation "Fig." should be when it appears in running text and should be followed by a number unless it comes at the beginning of a sentence, e.g.: "The results are depicted in Fig. 5. Figure 9 reveals that..." (see the "Manuscript composition" of Manuscript preparation guidelines for authors of AMT);**

We modified the abbreviation of the figure in the text of the revised version of the manuscript according to the AMT guidelines.

**e) The TBRi(Ttip) acronym is not commented in the text and in the figure legend;**

The terms SL3, LGO and $TBR_i$ indicate the three tipping bucket instruments here considered (the Shanghai SL3-1, the Logotronic and the synthetic/ideal TBRG respectively). The subscript indicates the algorithm used to obtain the RI values from the TBRG's data. The subscript "Ttip" denotes the use of the inter-tip algorithm, while the "raw" subscript is employed to denote the use of the tip counting algorithm. This point is now clarified in the legend of the figures and in the text of the revised manuscript.

**f) page 5, line 21, list the figure of this paragraph;**

This paragraph still refers to Fig. 4 (Fig. 5 in the revised version of the manuscript) like the previous one. Explicit reference is now added in the revised text.

**g) improve the Y axis scale of graphic of figure 5 (review all scales of all graphics).**

The scale of the Y axis (for both Figure 4 and 5) has been limited to [-100%, 100%] to allow appreciating both the large variability of the results below 6 $\mathrm{mmh^{-1}}$ and the changes above that value, even with some boxplots actually exceeding this range. A larger scale would compromise the readability of the second part of the graph.

**In the "5 Conclusions" section: - a) page 6, line 31, cite examples of sources of uncertainties in field operation;**

In field operation, instruments are subject to environmental factors like wind-induced effects or the presence of particles in the atmosphere and in the rainwater. Wind-induced effects can lead to an underestimation of the rain amount, which is composed by a systematic bias and an unknown uncertainty. A few examples are now cited in the concluding section.

**b) rewrite the paragraph to make the text clearer, once the rain intensity is calculated from the measured rain (tips) made by TBRGs and time resolution used is theusually 1 minute, but other time resolutions may be chosen. Make it clear when compare results obtained by different forms of RI calculations.**

In this study, we considered only the time resolution of one minute (as recommended by the WMO) for the calculation of the RI, for both the inter-tip and the tip-counting algorithms. A sensitivity analysis on the time resolution of the RI output would be of interest as well, although it is beyond the scope of the present paper.

**References**

JCGM: International vocabulary of metrology - Basic and general concept and associated terms (VIM), Joint Committee for Guidelines in Metrology, 3rd edn., 2012.

---

## Author Comment (AC2) · 25 Oct 2016

Matteo Colli1,2, Mattia Stagnaro1,2, Luca G. Lanza1,2, and P.W. Chan3

1University of Genova, Department of Civil, Chemical and Environmental Engineering, Via Montallegro 1, 16145 Genova, Italy

2WMO/CIMO Lead Centre "Benedetto Castelli" on Precipitation Intensity, Italy

3Hong Kong Observatory, 134A Nathan Road, Honk Kong, China

Correspondence to: Mattia Stagnaro (Mattia.Stagnaro@unige.it)

We would like to thank Prof. Remko Uijlenhoet for his helpful comments. In a separate comment, we present a revised version of the paper, which also includes corrections of the minor issues that have been pointed out by the reviewer. Below we provide a point-by-point reply to the issues raised in his review (reported in bold).

First of all, I am not convinced Fig. 6 (the Taylor diagram) is the best way to present the results of the intercomparison.

- 5 In any case, it was not easy for me to decipher it (particularly the choice of colors for the symbols was confusing to me). In our opinion, the Taylor diagram (reported here in Figure 1 for your convenience) is a useful representation to show in the same graph the standard deviation of the rainfall intensity signal, the RMS difference of each event from the reference and the correlation coefficient between the one-minute RI signal derived from each TBRG measurement and the reference one. From Fig. 1, it is possible to notice at the same time that, using the inter-tip instead of the tip-counting algorithm, improves
- 10 the correlation coefficients of the two TBRGs measures, reduce the RMS values with respect to the reference, but it has no significant effects on the standard deviation values (here expressed in terms of their normalized values).

Concerning the symbols, we used crosses to indicate the single event values while filled circles summarize the averaged values of all considered events. Colors are now changed from the initial version of the manuscript. The red values indicate the ideal TBRG and the black dot represents the OSK reference gauge.

15 Also, the "residual uncertainty in the calibration process" (see the Conclusions section) asks for further discussion. Are there ways to account for this remaining error source in the calibration process?

This is the calibration uncertainty, and it can be reduced - in general - by using accurate calibration facilities and standard methods for calibration that are traceable to the international standards. Unfortunately, no such standard exists at the international level for rain gauge calibration; therefore we usually follow the Italian national standard, dated 2012 and cited in the references

20 of the paper (a new standard is presently in preparation under CEN). The magnitude of the calibration uncertainty is included in the criterion used to assign a class to the instrument in the Italian standard (three classes depending on the residual bias and the associated uncertainty). Although this is recognized by the WMO, following the recent Laboratory and Field Intercomparison of Rain Intensity Gauges in 2005 and 2009, no international reference is yet available. This is why we did not mention it and just commented about the "residual uncertainty in the calibration process".

**Figure 1.** Taylor diagram representation of pattern statistics of the various RI series. The radial distance from the origin is proportional to the normalized standard deviation of the RI signal; the blue contour lines highlight the RMS difference from the OSK reference (black dot) for each recorded event; the azimuthal position indicates the correlation coefficient between the RI signal and the reference. Crosses indicate the statistics of each single event, while the dots indicate the average values of the whole campaign (colors according to the legend).

---

## Author Comment (AC3) · 25 Oct 2016

Matteo Colli[1,2], Mattia Stagnaro[1,2], Luca G. Lanza[1,2], and P.W. Chan[3]

[1]University of Genova, Department of Civil, Chemical and Environmental Engineering, Via Montallegro 1, 16145 Genova, Italy
[2]WMO/CIMO Lead Centre "Benedetto Castelli" on Precipitation Intensity, Italy
[3]Hong Kong Observatory, 134A Nathan Road, Honk Kong, China

*Correspondence to:* Mattia Stagnaro (Mattia.Stagnaro@unige.it)

We would like to thank referee #3 for his/her constructive comments. Below we report the referee comments in bold and our answer. A revised version of the manuscript, containing the reviewer's suggestion, is provided in the supplement of a separate comment.

**Abstract. I suggest to include in the abstract the numerical values of the RI interval where the impact of the algorithm is more relevant.**

Following the referee suggestion, we added the RI interval (6-50 $\mathrm{mmh^{-1}}$)where the inter-tip algorithm shows the most relevant effects.

**Section 2. The relative distance between the three instruments should be indicated here.**

Following the suggestion of referee #1 a brief description of the test field site has been added in the revised version of the manuscript, including the distance information (see comment c - Section 2 of the reply to Referee #1).

**Lines 22 and 26 (and throughout the paper), please be consistent with the labeling of the OSK data (OSK or DC or REF).**

The manuscript has been reviewed and the term OSK has been chosen to identify the drop-counter reference throughout the paper.

**Section 3. In equation 1 and 2 a summation notation should be used, and in equation 2 the second member should be multiplied by 100.**

Equation 1 represents the relative difference between the one-minute RI instrumental measurement ($RI_m$) and the reference one-minute RI value. This calculation has been performed for each minute of all the events here considered.

Equation 2 indicates the calculation of the standardized one-minute RI. No summation notation is needed in the formulas because they are used to calculate each one-minute value, rather than any integral (or average) quantity.

The request from the reviewer may arise from the use of the percentage notation in Eq. 2, which is a typo from the first draft of the manuscript and is now deleted from the revised version.

**Figure 7 is a 3-panel figure, please describe in the caption all panels and put labels accordingly.**

Figure 7 has been re-edited and the labels have been added to each panel of the figure. The lower panel (c) description has been added in the caption of the figure.

---

## Author Comment (AC4)

Matteo Colli[1,2], Mattia Stagnaro[1,2], Luca G. Lanza[1,2], and P.W. Chan[3]

[1]University of Genova, Department of Civil, Chemical and Environmental Engineering, Via Montallegro 1, 16145 Genova, Italy
[2]WMO/CIMO Lead Centre "Benedetto Castelli" on Precipitation Intensity, Italy
[3]Hong Kong Observatory, 134A Nathan Road, Honk Kong, China

*Correspondence to:* Mattia Stagnaro (Mattia.Stagnaro@unige.it)

We would like to thank referee #4 for his/her comments since they allowed us to improve the quality of the manuscript. Below we report the referee comments in bold and our answer. A revised version of the manuscript, containing the reviewer's suggestion, is provided in the supplement of a separate comment.

**In particular, the characteristics of the errors, as summarized by the relatively large observed biases and variabilities relative to the reference measurements, merits some discussion.**

We tried to summarize the discussion about the characteristics of both the bias and the variability of the observed RI output in the "Results" section of the manuscript. In particular, the descriptions of Fig. 5 and 6 precisely aim at providing the requested discussion. The same is true for the descriptions provided in the text with reference to Fig. 7 and 8. Consider also that the reference is here provided by a high-resolution optical gauge, but this has associated inaccuracies as well - which are totally unknown.

**One of the referees has commented on the need for a site description which should include the spatial layout/separation distances/mounting heights of the gauges etc that might influence the characteristics of the errors.**

The site description of the field test site have been added to the revised version of the paper following the suggestion of referees #1 and #3.

**Given that the TBRGs are stated in the paper to have been subjected to laboratory calibration/ correction of the intensity data using a correction curve, it does not seem that the errors observed in relation to the reference raingauge can be attributed to "instrumental mechanical errors", given that the correction curve is supposed to account largely for these effects.**

We added some statements in the discussion of the results (section 5) in order to stress that the variability of the errors after correcting for calibration are only partially attributable to the remaining calibration uncertainty, as commented by the Reviewer. We stated that there are other factors that contribute to the measurements accuracy such as the RI calculation method adopted (inter-tip vs. simple count) and environmental factors (e.g. the wind-induced and wetting effects).

**Small scale rainfall variability across the measurement site must make a contribution to the observed variability.**

The OSK reference is distant approximately 5.8 and 2.1 m from the SL3 and LGO rain gauges respectively. All the three rain gauges are installed on the ground in the test field site of the Hong Kong Observatory and they are placed far from relevant obstacles. Due to this configuration, it is possible to assume that the impact of rainfall spatial variability is negligible.

5     **In relation to the consistent underestimation observed, which should be largely removed by the correction curve, was a volumetric check gauge run alongside the two TBRGs and the reference gauge to check on the total volumes from each over the period of the experiment?**

The reviewer is correct to propose a volumetric check of the TBRGs to investigate the observed underestimation; however, data from co-located totalizer gauges were not available during the campaign. The objective of the study was not to assess the

10    best performing gauge, nor to achieve the most accurate measurements in absolute terms, but rather to compare two different algorithms for RI retrieval. In this view, a residual bias between the two TBRGs and the reference is not expected to affect the results, which are mainly based on variability analysis (standard deviations).

---

## Author Comment (AC5)

**Performance of post-processing algorithms for rainfall intensity using measurements from tipping-bucket rain gauges**

Mattia Stagnaro[1,2], Matteo Colli[1,2], Luca Giovanni Lanza[1,2], and Pak Wai Chan[3]

[1]University of Genova, Department of Civil, Chemical and Environmental Engineering, Via Montallegro 1, 16145 Genova, Italy
[2]WMO/CIMO Lead Centre "Benedetto Castelli" on Precipitation Intensity, Italy
[3]Hong Kong Observatory, 134A Nathan Road, Honk Kong, China

*Correspondence to:* Mattia Stagnaro (Mattia.Stagnaro@unige.it)

**Abstract.** Eight rain events recorded from May to September 2013 at the Hong Kong International Airport (HKIA) have been selected to investigate the performance of post-processing algorithms used to calculate the Rainfall Intensity (RI) from Tipping-Bucket Rain Gauges (TBRGs). We assumed a drop counter catching-type gauge as a working reference and compared rainfall intensity measurements with two calibrated TBRGs operated at a time resolution of 1 min. The two TBRGs differ in

5    their internal mechanics, one being a traditional single-layer dual-bucket assembly, while the other has two layers of buckets. The drop counter gauge operates at a time resolution of 10 s, while the time of tipping is recorded for the two TBRGs. The post-processing algorithms employed for the two TBRGs are based on the assumption that the tip volume is uniformly distributed over the inter-tip period. A series of data of an ideal TBRG is reconstructed using the virtual time of tipping derived from the drop counter data. From the comparison between the ideal gauge and the measurements from the two real TBRGs the

10    performance of different post-processing and correction algorithms are statistically evaluated over the set of recorded rain events. The improvement obtained by adopting the inter-tip time algorithm in the calculation of the RI is confirmed. However, by comparing the performance of the real and ideal TBRGs, the beneficial effect of the inter-tip algorithm is shown to be relevant for the mid-low range (6-50 $\mathrm{mmh^{-1}}$) of rainfall intensity values (where the sampling errors prevail), while its role vanishes with increasing the RI, in the range where the mechanical errors prevail.

**1   Introduction**

15    Application-driven requirements of rainfall data (see e.g. Lanza and Stagi, 2008), the recommendations of international bodies such as the World Meteorological Organization (WMO, 2008), and new measurement quality standards issued at the national (UNI 11452:2012; BS 7843-3:2012) and international (CEN/TR 16469:2013) level provide an increasingly demanding framework in terms of proven instrumental accuracy and reliability.

20    Following the effort leaded in the last decade by WMO and aimed at quantifying the achievable accuracy of rainfall intensity measurements (Lanza and Vuerich, 2009), both users and manufacturers of precipitation gauges are developing strategies to reduce the uncertainty and to provide suitably documented performance evaluation of rainfall measurements.

Sound metrological procedures for the assessment of the uncertainty of meteorological measurements have recently been

introduced within the framework of Europe-wide collaborative projects (Merlone et al., 2015), and therein extended to the measurement of liquid precipitation (see Santana et al., 2015). In this context, we use the term uncertainty in accordance with the International vocabulary of metrology (VIM) as the non-negative parameter characterizing the dispersion of the quantity values being attributed to a measurand (JCGM, 2012).

5 Beside the inherent instrumental factors (e.g. the systematic mechanical bias of tipping-bucket rain gauges, or the dynamic response bias of weighing gauges), post-processing of the raw data to obtain accurate rain intensity records at a pre-determined temporal resolution is common practice. In the case of tipping-bucket rain gauges (TBRGs), dedicated post-processing algorithms must be employed to achieve sufficient accuracy and to minimize the impact of sampling errors and the discrete nature of the measurement.

[revised manuscript text omitted]

For both the TBRGs, the two values of one-minute $RI$ derived from the two post-processing algorithms described above (generally indicated in Eq.(1) as the measured rain intensity $RI_m$) were employed to calculate the accuracy of rainfall intensity measurements in terms of deviations ($e_{rel}$) from the OSK drop-counter reference value (RI$_{ref}$) as follows:

$$e_{rel}(\%) = \frac{RI_m - RI_{ref}}{RI_{ref}} \cdot 100 \tag{1}$$

In addition, in order to compare the performance of post-processing and correction algorithms for the TBRG measurements, we derived a virtual sequence of tips of an ideal tipping-bucket rain gauge (TBR$_i$) from the high-resolution drop-counter (OSK) data. This simulates the behavior of a best performing TBRG, representing the maximum accuracy to be expected when using tipping-bucket mechanics. The inter-tip and the tip-counting algorithms were employed to derive the one-minute ideal rainfall intensity values ($RI_i$ and $RI_{i(raw)}$ respectively) and then the relative deviation ($e_{rel}$) from the reference (OSK) as described in Eq.(1).

In order to assess the capability of the employed algorithm to describe the inner variability of the considered events and to capture their finer details, we calculated the correlation coefficients between all the derived time series and the reference ones. In particular, for each event, we calculated the RMS error of paired deviations between the measured and the ideal/reference RI signal.

5     In an effort to homogenize the dataset, we considered the standardized value of the generic rainfall intensity value ($RI_n$) obtained as follow:

$$RI_n = \frac{(RI_m - \mu_{ref})}{\sigma_{ref}} \tag{2}$$

where the mean value $\mu_{ref}$ and the standard deviation $\sigma_{ref}$ employed in Eq.(2) to obtain the standardized values of rainfall intensity ($RI_n$) for all the TBRGs, are those derived from the OSK drop counter reference rainfall rates. In this way, we
10   obtained comparable results in terms of standard deviation of the standardized RI time series for all the investigated rainfall events.

**4   Results**

We first evaluated the accuracy of the investigated RI algorithms using TBRGs measurements by comparing their performance with the working reference. Figure 5 shows the relative deviations ($e_{rel}$) from the drop-counter (OSK) reference for the three
15   TBRGs (including the ideal virtual gauge). The reported boxplots provide a synthesis of the results obtained adopting the inter-tip algorithm, which are classified according to different ranges of $RI_{ref}$.

Two different regions of this graph show different behavior of the relative deviations calculated with the inter-tip approach: for low values of the $RI$ the relative deviations of all the TBRGs exhibit a large variability, whereas this scatter suddenly decreases just above the $RI$ value of 6 $\mathrm{mm\ h^{-1}}$ (highlighted in black) and then continues to reduce with increasing the $RI$. This limit
20   coincides with the sensitivity of the TBRGs buckets (both the real and virtual ones); in fact, values of RI higher than 6 mm/h generate at least one tip per minute for TBRGs with a sensitivity of 0.1 $\mathrm{mm}$.

It emerges from the graph in Fig. 5 that the average deviation of both the LGO and SL3 gauges is always negative. In particular, the SL3 gauge shows an average underestimation that is larger than the LGO (except for one bin). This is coherent with the daily amounts shown in Fig. 4, where the cumulative rain depth of the SL3 gauge is lower than or equal to the value recorded
25   by the LGO. Despite the fact that the SL3 rain gauge underestimates rainfall more than the LGO on average, the variability of the deviations from the reference for different values of RI is slightly reduced with respect to the LGO.

The behavior of the ideal TBRG is clearly different, since it is not affected by instrumental mechanical errors (ideal mechanics), and the average value of $e_{rel}$ becomes close to zero immediately after the threshold value of the instrument sensitivity.

In Fig. 5 it can be noted that the ideal TBRG shows a large variability and an average value of the relative deviations that is
30   comparable to the real TBRGs when the RI values are below 6 $\mathrm{mm\ h^{-1}}$. This means that the error caused by the aggregation time is relevant in this region when compared to the mechanical one.

As the rainfall rate increases, the variability of $e_{rel}$ considerably decreases, even if does not vanish due to the sampling time

of the OSK (10 seconds), whereas the average values are close to zero. The residual variability of $e_{rel}$ observed for the real TBRGs accounts for the propagation of the calibration uncertainty and other environmental factors (e.g. the wind-induced and wetting effects).

Figure 6 describes the relative deviation ($e_{rel}$) from the drop-counter OSK reference of the measured value of RI from the three TBRGs adopting the rough tip counting algorithm. Below the instrument sensitivity limit of 6 mm h$^{-1}$, where less than one tip per minute occurs, the relative errors of all TBRGs are similar and show a very high scatter. Above this threshold value, the variability gradually decreases. Also in this case the ideal TBRG shows an average value of deviation which is close to zero, while the SL3 and LGO continue to maintain a mean value of $e_{rel}$ negative for all the RI classes . Close to the threshold value of 6 mm h$^{-1}$, the variability of the relative error ($e_{rel}$) is drastically reduced for all three gauges, and the average value is the closest to zero.The residual variability of e$_{rel}$ observed for the real TBRGs when RI $> 6$ mmh$^{-1}$ accounts for the residual tipping bucket mechanical error after calibration and other environmental factors (e.g. the wind-induced and wetting effects).

[revised manuscript text omitted]

JCGM: International vocabulary of metrology - Basic and general concept and associated terms (VIM), Joint Committee for Guidelines in Metrology, 3rd edn., 2012.

Lanza, L. and Stagi, L.: Certified accuracy of rainfall data as a standard requirement in scientific investigations, Advances in geosciences, 16, 43–48, 2008.

Lanza, L. G. and Stagi, L.: High resolution performance of catching type rain gauges from the laboratory phase of the WMO Field Intercomparison of Rain Intensity Gauges, Atmospheric Research, 94, 555–563, 2009.

Lanza, L. G. and Vuerich, E.: The WMO field intercomparison of rain intensity gauges, Atmospheric Research, 94, 534–543, 2009.

Merlone, A., Lopardo, G., Sanna, F., Bell, S., Benyon, R., Bergerud, R., Bertiglia, F., Bojkovski, J., Böse, N., Brunet, M., et al.: The MeteoMet project–metrology for meteorology: challenges and results, Meteorological Applications, 22, 820–829, 2015.

Santana, M. A., Guimarães, P. L., Lanza, L. G., and Vuerich, E.: Metrological analysis of a gravimetric calibration system for tipping-bucket rain gauges, Meteorological Applications, 22, 879–885, 2015.

Taylor, K. E.: Summarizing multiple aspects of model performance in a single diagram, Journal of Geophysical Research, 106, 7183–7192, 2001.

UNI 11452:2012: Hydrometry - Measurement of rainfall intensity (liquid precipitation) - Metrological requirements and test methods for catching type gauges., Standard, Ente Nazionale Italiano di Unificazione, Milano, IT, 2012.

WMO: Guide to Meteorological Instruments and Methods of Observation-No. 8, World Meteorological Organization, 7th edn., 2008.

[Figure]

**Figure 1.** Shanghai SL3-1 (SL3): internal mechanism with double layer of tipping buckets (a) and external case (b).

[Figure]

**Figure 2.** The West corner of the Hong Kong Observatory field test site where the SL3 (blue box), the LGO (red box) and the OSK (green box) are located. The distance of each instrument from the field site borders is indicated.

[Figure]

**Figure 3.** Reference rainfall intensity measured by the OSK drop counter (shaded gray background) and comparison of the accumulated reference (red continuous line) with the accumulated Logotronic (LGO) and Shanghai (SL3) measurements during the sample event of 22 May 2013.

[Figure]

**Figure 4.** Box plot of the reference rainfall intensity for each event (top of the graph) and corresponding daily rain amount (lower part of the graph) for the OSK drop counter (reference) and the two TBRGs (SL3 and LGO). The explanation of the symbols used in the box-plot representation is shown on the right side of the graph.

[Figure]

**Figure 5.** One-minute relative deviations ($e_{rel}$) between the three TBRGs (including the ideal one) and the reference (OSK) when adopting the inter-tip post-processing algorithm (Ttip).

[Figure]

**Figure 6.** One-minute relative deviations ($e_{rel}$) between the three TBRGs (including the ideal one) and the reference (OSK) when adopting the tip counting algorithm (raw).

[Figure]

**Figure 7.** Taylor diagram representation of pattern statistics of the various RI series. The radial distance from the origin is proportional to the normalized standard deviation of the RI signal; the blue contour lines highlight the RMS difference from the OSK reference (black dot) for each recorded event; the azimuthal position indicates the correlation coefficient between the RI signal and the reference. Crosses indicate the statistics of each single event, while the dots indicate the average values of the whole campaign (colors according to the legend).

[Figure]

**Figure 8.** Standard deviation of the relative error for the ideal and the investigated TBRGs when adopting the inter-tip time algorithm (Ttip) and the tip counting method (raw) with respect to the ideal gauge for both the SL3 (a) ad LGO (b) instruments. In panel (c) the sample size for each RI class is reported.

**Table 1.** Type of rain gauges employed for this comparison and their principal characteristics.

| Rain gauge | Measuring principles | Resolution (mm) | Max. RI* (mm h$^{-1}$) | Funnel diameter (mm) | Accuracy (%) |
|---|---|---|---|---|---|
| Ogawa OSK PC1122 | Drop counter | $5.21 \cdot 10^{-3}$ | 200 | 127.0 | $\pm 2.89$ |
| Logotronic (LGO) | Tipping-bucket | 0.1 | 200 | 252.3 | $\pm 2$** |
| Shanghai (SL3) | 2-layers tipping-bucket | 0.1 | 240 | 200.0 | $\pm 0.4$** |

* Maximum measured rainfall intensity as declared by the manufacturer.

** Accuracy provided by the manufacturer.

**Table 2.** Total rainfall depth ($h_{tot}$), maximum one-minute rainfall rate ($I_{max}$) and duration ($d$) of selected events recorded by the Ogawa drop counter during the observation period.

| Date | $h_{tot}$ (mm) | $I_{max}$ (mm h$^{-1}$) | $d$ |
|---|---|---|---|
| 22 May 2013 | 205 | 125 | 11h 27' |
| 25 May 2013 | 125 | 102 | 06h 56' |
| 24 June 2013 | 142 | 114 | 06h 19' |
| 24 July 2013 | 51 | 95 | 02h 54' |
| 25 July 2013 | 48 | 96 | 03h 10' |
| 04 September 2013 | 66 | 72 | 06h 44' |
| 05 September 2013 | 65 | 73 | 04h 53' |
| 23 September 2013 | 89 | 64 | 06h 09' |